# Bladder Cancer Metastasis Induced by Chronic Everolimus Application Can Be Counteracted by Sulforaphane In Vitro

**DOI:** 10.3390/ijms21155582

**Published:** 2020-08-04

**Authors:** Saira Justin, Jochen Rutz, Sebastian Maxeiner, Felix K.-H. Chun, Eva Juengel, Roman A. Blaheta

**Affiliations:** Department of Urology, Goethe-University, 60323 Frankfurt am Main, Germany; justinsaira@hotmail.com (S.J.); Jochen.Rutz@kgu.de (J.R.); Sebastian.Maxeiner@kgu.de (S.M.); Felix.Chun@kgu.de (F.K.-H.C.); Eva.Juengel@unimedizin-mainz.de (E.J.)

**Keywords:** sulforaphane, bladder cancer, adhesion, chemotaxis, integrins, CD44

## Abstract

Chronic treatment with the mTOR inhibitor, everolimus, fails long-term in preventing tumor growth and dissemination in cancer patients. Thus, patients experiencing treatment resistance seek complementary measures, hoping to improve therapeutic efficacy. This study investigated metastatic characteristics of bladder carcinoma cells exposed to everolimus combined with the isothiocyanate sulforaphane (SFN), which has been shown to exert cancer inhibiting properties. RT112, UMUC3, or TCCSUP bladder carcinoma cells were exposed short- (24 h) or long-term (8 weeks) to everolimus (0.5 nM) or SFN (2.5 µM), alone or in combination. Adhesion and chemotaxis along with profiling details of CD44 receptor variants (v) and integrin α and β subtypes were evaluated. The functional impact of CD44 and integrins was explored by blocking studies and siRNA knock-down. Long-term exposure to everolimus enhanced chemotactic activity, whereas long-term exposure to SFN or the SFN-everolimus combination diminished chemotaxis. CD44v4 and v7 increased on RT112 cells following exposure to SFN or SFN-everolimus. Up-regulation of the integrins α6, αV, and β1 and down-regulation of β4 that was present with everolimus alone could be prevented by combining SFN and everolimus. Down-regulation of αV, β1, and β4 reduced chemotactic activity, whereas knock-down of CD44 correlated with enhanced chemotaxis. SFN could, therefore, inhibit resistance-related tumor dissemination during everolimus-based bladder cancer treatment.

## 1. Introduction

Bladder cancer is the fourth most common type of cancer in men, ninth most common in women, and is ranked as the most common urological malignancy in the majority of countries. Approximately 25% of newly diagnosed bladder cancers are already classified as muscle invasive (stage ≥ T2), and progress to muscle invasion in initially non-muscle-invasive bladder cancer in about 10–20% of cases [1] occurs. The prognosis of patients with metastatic disease is poor, with overall survival approximating only 15% [2].

Genome-based subtyping has revealed many oncogenic targets contributing to bladder cancer development, e.g., fibroblast growth factor receptor (FGFR), PI3K (phosphoinositid-3 kinase), Akt (serine/threonine kinase), and mTOR (mechanistic target of rapamycin) [3]. Abnormal activation of these pathways leads to survival and proliferation of tumor cells along with metastatic competence, angiogenesis, and therapy resistance [4]. One way of tackling bladder cancer is to inhibit mTOR, where anomalous interconnected signaling pathways converge [5].

Currently, everolimus is the only mTOR inhibitor approved by the U.S. Food and Drug Administration to treat breast cancer, advanced renal carcinoma, pancreatic neuroendocrine tumor and subependymal giant-cell astrocytoma, thus opening this therapeutic approach for bladder carcinoma as well [6]. mTOR inhibitors are cutting-edge agents for targeted treatment but are still, as a monotherapy, not that efficient. Like any other antineoplastic strategy, long-term everolimus treatment causes resistance initiated by genomic instability, though the underlying mechanisms are still unclear [7]. Chemoresistance, along with severe side effects under conventional treatment, drive many cancer patients to complementary or alternative medicine (CAM) treatment options. Indeed, integrative medicine, especially green chemoprevention, has become very popular, particularly for patients with advanced tumors and a poor prognosis with conventional therapy [8,9].

Isothiocyanate sulforaphane (SFN) is the hydrolytic product of glucosinolates, found in cruciferous vegetables, such as broccoli, cabbage, and kale. SFN acts as a natural histone deacetylase (HDAC) inhibitor. This is noteworthy, since HDAC serves as a critical epigenetic regulator in bladder cancer and is closely involved in metastatic progression [10]. Hence, HDAC inhibition has been introduced as a therapeutic target due to its role in regulating cell growth, invasion, and apoptosis [11]. Several phytochemicals have meanwhile been identified as “epi-drugs” by altering epigenetic modification in multiple cancer types [12], making natural HDAC inhibitors relevant for clinical use.

SFN, as one of these natural HDAC inhibitors, has been reported to exert both chemo-preventive and anti-carcinogenic effects on various tumor types [13]. It is of particular interest that combining HDAC and mTOR inhibitors may not only elicit additive effects but also counteract resistance induction towards an mTOR inhibitor based regimen. Whether this holds true for SFN in treating bladder cancer has not yet been evaluated. Therefore, the present study was designed to investigate metastatic characteristics of bladder carcinoma under short- and long-term SFN treatment, combined with the mTOR-inhibitor, everolimus.

## 2. Results

### 2.1. Cell Adhesion

#### Acute and Chronic Treatment

Adhesion of RT112 was inhibited with everolimus, while SFN or the SFN-everolimus combination elevated RT112 adhesion. UMUC3 and TCCSUP cells responded differently. Here, adhesion decreased with SFN or the SFN-everolimus combination (UMUC3) (Figure 1, upper panels).

Chronic everolimus exposure increased RT112 adhesion, in contrast to the inhibiting effect of acute everolimus treatment. UMUC3 and TCCSUP adhesion was diminished, whether the cells were chronically exposed to SFN or everolimus alone or in combination (Figure 1, lower panels).

### 2.2. Chemotactic Activity under Acute Versus Chronic Drug Application

Acute everolimus exposure in RT112 cells significantly decreased motility, whereas the opposite effect was apparent with chronic exposure to everolimus. The same response was seen with the TCCSUP model. No sign of everolimus resistance was seen in UMUC3 cells, since both acute and chronic treatment resulted in less transmigration (Figure 2). SFN alone blocked chemotaxis in all three cell lines, even after chronic exposure. The SFN-everolimus combination also suppressed chemotactic movement, counteracting the undesired increase observed with chronic everolimus exposure in RT112 and TCCSUP cells (Figure 2). The everolimus-SFN combination was superior to the chronic application of everolimus alone in UMUC3 cells. Since the resistance phenomenon was most obvious in RT112 cells, as demonstrated by strong chemotaxis activation with chronic everolimus exposure, this cell line was used in subsequent experiments.

### 2.3. Surface Expression of CD44 Splice Variants

CD44v3, v4, v5, v6, and v7 were all expressed on the RT112 cell surface (Figure 3A). Acute treatment with everolimus or SFN or the drug combination did not significantly alter the CD44v expression level, except for CD44v7, which was up-regulated with combined SFN + everolimus (Figure 3B). Distinct modifications were seen with chronic treatment. Everolimus alone moderately elevated CD44v4 and v7, whereas SFN alone enhanced all CD44 variants, with the strongest effects exerted on CD44v4. Combined drug use resulted in additive effects, particularly evident with CD44v4 and v7 expression (Figure 3B).

### 2.4. CD44 Knock-down

To investigate the relevance of CD44 on adhesion and chemotaxis, CD44 was knocked down in RT112 cells that had not been treated with SFN or everolimus and adhesion and chemotaxis were then evaluated. The successful knock-down of CD44 (Figure 4A) only slightly altered adhesion, but a strong increase in chemotaxis was apparent (Figure 4B).

### 2.5. Surface and Protein Expression of Integrin Subtypes

The integrin members α2, α3, α6, αV, and β1 were strongly expressed on RT112 cells. Integrins α5 and β4 were weakly expressed (Figure 5A), while α1, α4, and β3 were not expressed at all (data not shown). Acute treatment with everolimus or the everolimus-SFN combination increased α3, αV, and β1, whereas SFN alone had no influence on these integrin subtypes (Figure 5B). Chronic everolimus application led to an increase in all analyzed integrin subtypes, except for α2, which remained unaltered (Figure 5B). SFN treatment caused an increase in α3, α5, and αV but significantly diminished β4. Combined SFN-everolimus up-regulated integrin α3 in an additive manner. Integrin α5 was also elevated. The drug combination more strongly suppressed β4, compared to SFN alone.

In regard to the integrin protein analysis, only moderate responses were seen with acute drug treatment. Everolimus enhanced integrin αV and reduced β1, and SFN diminished β1 (Figure 6A). Profound modifications were evoked by chronic everolimus treatment. This was evidenced by an up-regulation of α2 (slightly), α5, α6, β1, and β4 and a down-regulation of α3. SFN enhanced α2 (slightly), α5, and β1 and reduced α3 and β4. Combined drug use resulted in strongly elevated α5 and profound inhibition of α3, αV, β1, and β4 (Figure 6A,B).

### 2.6. Integrin Blockade

Based on FACS analysis and western blots showing the integrin proteins most markedly affected by drug exposure, the integrins α3, α5, αV, β1, and β4 were blocked using function associated monoclonal antibodies and then adhesion and chemotaxis were evaluated. Blocking αV, β1, or β4 inhibited both adhesion and chemotaxis of RT112 cells. Blocking α3 or α5 enhanced chemotactic movement but did not alter adhesive behavior (Figure 6C).

## 3. Discussion

Acquired resistance to everolimus was particularly evident in the RT112 model, as demonstrated by reduced adhesion and chemotaxis with acute exposure but enhanced adhesion and invasion with long-term exposure. The resistance in these chronically exposed cells could be overcome by the natural HDAC inhibitor SFN. In other cell types resistance was manifested differently. In TCCSUP cells chemotaxis was enhanced but adhesion was down-regulated following chronic everolimus exposure. The enhanced invasive activity of RT112 cells could be a direct consequence of an enhanced number of cells contacting the collagen matrix, increasing the likelihood of migratory events. Elevated invasion of TCCSUP could be due to diminished firm tumor cell adhesion, allowing more cells to become motile and invasive. Chemotaxis of UMUC3 remained suppressed, even after chronic everolimus administration. Here the eight-week drug application may have been too short to induce distinct signs of resistance. It has been shown that incubating UMUC3 cells with increasing concentrations of the mTOR-inhibitor, temsirolimus, over six months does result in resistance, exhibited by reactivated tumor growth and proliferation with mTOR and mTOR-related protein activation [14].

Since RT112 cells exhibited the most pronounced resistance to everolimus, this cell line was examined in more detail. SFN alone and the SFN-everolimus combination reversed the long-term effect of everolimus in regard to increased invasion. The effect exerted by SFN, therefore, seems to dominate the effect exerted by everolimus, making SFN interesting for clinical use. Since SFN alone blocks cell invasion, even after long-term application, the compound itself may not induce resistance. Whether this holds true for longer time periods cannot be assessed since chronic exposure was limited to eight weeks in the present study. However, evidence has recently been presented that pancreatic cancer cells do not develop resistance to SFN, even after continuous exposure for more than one year [15]. SFN application could therefore also provide long-term benefit in treating bladder cancer.

Alteration in CD44 splice variants was analyzed since they are closely involved in cell-cell and cell-matrix interaction. Acute exposure to SFN or everolimus did not induce any modifications, except for CD44v7, which was up-regulated with the drug combination. However, radical changes occurred after eight weeks exposure. SFN enhanced the expression level of all CD44 variants, most prominently CD44v4 and CD44v7. The direction of change was not homogeneous, since everolimus induced CD44v4 and CD44v7 up-regulation, with additive effects in the presence of combined SFN/everolimus. Chronic everolimus exposure elevated chemotaxis, whereas SFN or the SFN-everolimus combination reduced it. These differences should be considered when discussing alterations of CD44v4 and CD44v7. Everolimus resistance has been characterized by elevated Akt-mTOR signaling [16]. Cross-communication between CD44 and Akt has recently been observed in drug resistant tumor cells, whereby enhanced CD44 expression is associated with Akt activation, driving tumor invasion forward [17,18]. Increased CD44v4 and CD44v7 expression induced by long-term exposure to the mTOR inhibitor everolimus may therefore serve to activate Akt, with Akt dependent signaling processes finally triggering enhanced motility.

This mechanism is not transferable to SFN, since chemotaxis was reduced in its presence. Indeed, upregulation of the CD44 variant v6 caused by SFN resulted in the inhibition of tumor cell invasion [19]. It has been shown that SFN suppresses Akt, along with the focal adhesion kinase (FAK) signaling pathways [20,21]. Interestingly, integrin β4, which interacts with FAK [22], was suppressed by SFN as well, but not by everolimus. Therefore, the increase of CD44v4 and CD44v7 induced by SFN could be coupled to the down-regulation of FAK and Akt signaling, finally preventing motile crawling of the tumor cells (via β4—see below). Nevertheless, Akt deserves further investigation. A recent investigation shows that the influence of SFN on mTOR-Akt signaling in bladder cancer leads to a significant increase in Akt phosphorylation (pAkt) in tumor cells following chronic everolimus administration [23]. Surprisingly, enhanced pAkt was also seen in the presence of SFN alone, but not in the presence of the everolimus-SFN combination where pAkt was reduced, compared to the untreated controls. The mTOR complex pRictor, associated with cell motility, was also investigated and found to be elevated by everolimus but diminished by SFN and the SFN-everolimus combination. Therefore, it is possible that the process of tumor cell invasion may also be suppressed by SFN via pRictor. Whether the elevation of pAkt in the presence of SFN indicates early signs of resistance caused by a feedback mechanism is not yet clear.

Speculatively, SFN may mitigate everolimus resistance by switching the function of CD44v4 and v7 from tumor promotion to tumor suppression. Bioinformatic analysis of cancer patients using The Cancer Genome Atlas database has revealed that production of CD44v splice variants by an epigenetic mechanism, including HDAC inhibition, may sustain an epithelial phenotype and prevent epithelial-mesenchymal transition [24], thus corroborating this speculation. Still, the role of the CD44 variants, v4 and v7, in cancer progression has not been evaluated in detail and requires further investigation.

Long-term everolimus exposure was also accompanied by alteration of the integrin α and β expression levels, compared to the integrin expression pattern in tumor cells acutely exposed to everolimus. α3 was massively up-regulated and α5 de novo enhanced, independent of the therapeutic regimen. Since protein analysis demonstrated intracellular reduction of α3, α3 was probably translocated from the cytoplasm to the outer cell surface. α5, however, remained intracellularly high, due to de novo synthesis. The integrin subtypes α6, αV, β1, and β4 were also significantly increased under chronic everolimus exposure. α6 has been shown to induce cell motility and invasion and promote drug resistance [25,26] and might, therefore, be partially responsible for reactivating the tumor cell invasion cascade by everolimus. This integrin subtype was not further investigated, since in RT112 cells, it was not as strongly influenced by drug exposure as were other integrin subtypes. However, integrin blocking studies done with αV, β1, and β4 monoclonal antibodies demonstrated a positive correlation between the αV, β1, and β4 surface expression level and tumor cell migratory activity. Therefore, everolimus resistance reflected by enhanced tumor cell chemotaxis might also be caused by up-regulated αV, β1, and β4. Up-regulation of αV or β1 did not occur in the presence of SFN (β1) or the SFN-everolimus combination (αV, β1). The relevance of integrin αV in bladder cancer has not been explored in detail. However, overexpression of this molecule in tumor tissue has been associated with poor, relapse-free survival of gastric and breast cancer patients, [27,28].

Likewise, high expression of the integrin β1 subtype positively correlated with tumor grade and poor clinical outcome in bladder cancer patients [29,30]. Prevention of αV, α6, and β1 up-regulation by SFN or the SFN-everolimus combination could, therefore, immobilize tumor cells to impede cancer progression. Down-regulation of β4 by SFN and, even more so by SFN combined with everolimus, indicates that β4, which mediates pro-migratory and pro-proliferative properties [31,32], could serve as a potential therapeutic target [33]. Since the combination with everolimus was superior to SFN alone, SFN may re-sensitize tumor cells to everolimus, similar to it re-sensitizing tumor cells to tamoxifen or lapatinib [34,35]. Prevention of α6, αV, and β1 elevation by SFN, along with diminishing the β4 integrin subtype could, therefore, make SFN a potent natural epi-drug to counteract the undesired long-term effects of everolimus.

Everolimus-induced resistance was manifested by increased chemotaxis and increased α3 and α5 in RT112 cells. This stood in direct contrast to the functional blockade of α3 and α5 that resulted in increased, instead of decreased, chemotaxis. Therefore, the induction of metastasis cannot be attributed to a discrete integrin related process. Rather, alteration in a set of several integrin subtypes, rather than a single subtype, may be necessary to induce resistance. Here, activation of the motile machinery in RT112 cells, as a result of α6, αV, β1, and β4 elevation, may dominate suppressive processes associated with elevated α3 and α5. In support, integrin blockage revealed no influence of α3 and α5 on adhesion, pointing to (at least) αV, β1, and β4 as the relevant drivers of resistance in this tumor type.

Investigative results presented here are related to the bladder cancer cell line RT112 and may not be generalizable. Similarities between everolimus-resistant RT112 and TCCSUP cells were seen in as much as SFN elevated α3, αv, and β1 in both cell lines. The α5 subtype was elevated in both cell lines by SFN as well. However, this was only significant in TCCSUP cells. In contrast, a different response was evoked in UMUC3 cells. Here, αv was down-regulated by SFN and the integrins α3 and β1 were not altered (Appendix A). The disparate mechanistic influence of SFN on different bladder cancer cell lines is not surprising. It has previously been shown that the HDAC inhibitor valproic acid suppresses adhesion in a broad panel of bladder cancer cell lines by altering both integrin α and β expression differently [36]. The molecular response of a particular tumor subline seems to depend on the integrin receptor set present on the cell surface. Integrin β4 is present on TCCSUP and RT112 but not on UMUC3 cells, whereas β3 has been detected on UMUC3 but not on TCCSUP and RT112 (Appendix A). Consequently, it may be expected that SFN treatment influences integrin subfamilies and integrin-related signaling in different cell lines differently. Indeed, differing integrin-guided adhesive behavior in several tumor sublines has been reported. Blocking the α3 integrin subunit has been shown to inhibit HCV29 bladder cancer cell attachment to the matrix proteins laminin and fibronectin but to exert an opposite effect on T24 and Hu456 cell adhesion. Similarly, blocking α5 integrin has been shown to down-regulate HCV29 and BC3726 cell-matrix interaction, whereas binding of the bladder cancer cell lines T24 and Hu456 is enhanced [37]. Based on the current data, it may be assumed that SFN acts on a set of integrin receptors, whereby the integrins modified by SFN may differ according to the initial characteristic integrin composition of the particular cell type.

High expectations are set on SFN serving as a novel anti-tumor agent [38]. However, questions do remain. Further studies about SFN metabolism and bioactivity are required. Bioavailability needs to be optimized, possibly by nanotechnologic approaches, to enhance drug delivery. Possible negative side effects or undesired interactions with well-established therapeutic regimens must also receive further evaluation. In vivo studies in animal models are the next step to advance knowledge about preventing resistance by administering SFN.

## 4. Materials and Methods

### 4.1. Cell Cultures

Three bladder carcinoma cell lines, RT112, UMUC3 (ATCC/LGC Promochem GmbH, Wesel, Germany), and TCCSUP (DSMZ, Braunschweig, Germany), were cultured in RPMI 1640 medium (Gibco/Invitrogen, Karlsruhe, Germany) supplemented with 10% fetal bovine serum (FBS), 2% HEPES buffer, 1% GlutaMAX, and 1% penicillin/streptomycin (all: Gibco/Invitrogen). RT112 cells represent a pathological stage T2, moderately differentiated, grade 2/3 tumor, UMUC3 an invasive high grade 3, and TCCSUP a grade 4 transitional cell carcinoma. Incubation was carried out at 37 °C in a humidified incubator with 5% CO_2_.

### 4.2. Drug Application

Everolimus (Novartis Pharma AG, Basel, Switzerland) was dissolved in dimethyl sulfoxide (DMSO) as a 10 mM stock solution and stored in aliquots at −20 °C. Prior to experiments, everolimus was diluted in cell culture medium. Ready-to-use L-sulforaphane was provided by Biomol, Hamburg, Germany (CAS registry number: CAS 142825-10-3). Dose response analysis showed that 0.5 nM everolimus and 2.5 µM SFN were optimal and these drug concentrations were employed. Untreated tumor cells served as controls. Pilot experiments demonstrated that 8 weeks chronic everolimus exposure leads to resistance development [39]. Tumor cells were, therefore, subjected to either short-term (24 h; specified as acute treatment) or long-term (8 weeks; specified as chronic treatment) application of SFN or everolimus alone, or to a SFN-everolimus combination for comparative analysis.

### 4.3. Cell Adhesion to Collagen Matrix

To evaluate tumor cell binding to immobilized matrix proteins, 24-well multi-plates (Falcon Primaria; Corning, Wiesbaden, Germany) were coated with 400 µg/mL of collagen G (Biochrom, Berlin, Germany) overnight at 4 °C. Plastic dishes served as background control. Plates were washed with 1% bovine serum albumin (BSA; Gibco/Invitrogen) in PBS (Gibco/Invitrogen) to prevent nonspecific cell adhesion. 1 × 10^5^ tumor cells were then added to each well for 1 h at 37 °C. Non-adherent tumor cells were subsequently washed off, and remaining adherent cells were fixed using 1% glutaraldehyde (Sigma, München, Germany). The evaluation was done microscopically by counting five different fields (each 0.25 mm²) with a raster ocular at 200-fold magnification. The mean cellular adhesion rate, defined by adherent cells_coated well_ - adherent cells_background_, was calculated.

### 4.4. Chemotaxis and Migration

Serum-induced chemotactic movement was investigated using a Boyden double chamber system with filters having 8 µm pores (Greiner Bio-One, Frickenhausen, Germany). 0.5 × 10^6^ cells/mL were placed in the upper chamber with serum-free medium. The lower chamber contained 10% FBS. Cells were incubated for 24 h and then fixed. The upper surface of the trans-well membrane was gently wiped using a cotton swab to remove non-migrating cells. Cells, which had moved to the lower surface of the membrane, were stained using hematoxylin (Sigma) and counted under a microscope at 200-fold magnification. The mean chemotaxis rate was calculated from five different observation fields (5 × 0.25 mm^2^).

### 4.5. CD44 Expression Analysis

A Lightning-Link Allophycocyanin (APC) Conjugation Kit was used to conjugate CD44 variants 44v3-7 antibodies according to the manufacturer’s instructions (eBioscience, ThermoFisher, Darmstadt, Germany). Cancer cells were detached and washed with blocking solution (PBS, 0.5% BSA). The cells were then incubated for 1 h at 4 °C with 2.5 µL APC conjugated monoclonal antibody directed against the following CD44 variants: anti-CD44v3 (clone VFF-327v3), anti-CD44v4 (clone VFF-11), anti-CD44v5 (clone VFF-8), anti-CD44v6 (clone VFF-7), and anti-CD44v7 (clone VFF-9; all: Bio-Rad, Feldkirchen, Germany ). 5 µl APC mouse IgG1, K (clone P3.6.2.8.1; ThermoFisher, Dreieich, Germany) served as the control isotype. CD44 expression was then measured using a FACscan (BD Biosciences; FL4-H (log) channel histogram analysis; 1 × 10^4^ cells per scan) and expressed as mean fluorescence units (MFU).

### 4.6. Integrin Surface Expression

Cancer cells were detached using accutase (PAA Laboratories GmbH, Pasching, Austria) and washed with blocking solution (PBS, 0.5% BSA). The cells were then incubated for 1 h at 4 °C with 20 µL ready-to-use phycoerythrin (PE) conjugated monoclonal antibody, directed against the following integrin subtypes: anti-α2 (IgG2a; clone 12F1-H6, 20 µL), anti- α3 (IgG1; clone C3II.1, 20 µL), anti- α5 (IgG1; clone IIA1, 20 µL), anti- α6 (IgG2a; clone GoH3, 20 µL), anti-β1 (IgG1; clone MAR4, 20 µL), anti- β 3 (IgG1; clone VI-PL2, 20 µL), or anti- β4 (IgG2a; clone 439-9B, 20 uL; all: BD Biosciences). 10 µL of anti-αV (IgG1; clone 13C2, Abcam, Cambridge, UK) was also used according to the manufacturer’s instructions. The integrin expression of tumor cells was then measured using a FACscan (BD Biosciences; FL2-H (log) channel histogram analysis; 1 × 10^4^ cells per scan) and expressed as mean fluorescence units. Mouse IgG1-PE (MOPC-21), mouse IgG2a-PE (G155-178) or rat IgG2b-PE (R35-38; all: BD Biosciences) was used as isotype control.

### 4.7. Western Blot Analysis

To visualize and quantify the integrin proteins, RT112 cell lysates were applied to a 7% polyacrylamide gel and electrophoresed for 90 min at 100 V. The proteins were then transferred to nitrocellulose membranes (1 h, 100 V). After this, membranes were blocked with non-fat dry milk for 1 h and incubated overnight with monoclonal antibodies directed against the following integrin proteins: anti-α2 (clone 2, 1:250), anti- α3 (1:500; both: Merck Millipore, Darmstadt, Germany), anti- α5 (clone 1, 1:5000), anti- αV (clone 21/CD51, 1:250), anti-β1 (clone 18, 1:2500), anti- β3 (clone 1, 1:2500), or anti- β4 (clone 7, 1:250; all: BD Biosciences) and anti- α6 (dilution 1:1000, Cell Signaling, Frankfurt, Germany). HRP-conjugated goat anti-mouse IgG and HRP-conjugated goat anti-rabbit IgG (both: Cell Signaling) served as the secondary antibodies. Membranes were briefly incubated with chemiluminescence (ECL) detection reagent (Amersham/GE Healthcare, München, Germany) to visualize the proteins and then analyzed using the Fusion FX7 system (Peqlab, Erlangen, Germany). β-actin (Cell Signaling) served as the internal control. GIMP 2.8 software was used to analyze the pixel density of the protein bands and to calculate the ratio of protein intensity/β-actin intensity.

### 4.8. Integrin Receptor Blockade

Tumor cells were incubated for 60 min with 10 μg/mL function-blocking anti-integrin α3 (clone P1B5), anti-integrin α5 (clone P1D6), anti-integrin αV (clone AV1), anti-integrin β1 (clone 6S6), or anti-integrin β4 (clone ASC-8) mouse mAb (all from Merck Millipore). Subsequently, tumor cell adhesion and chemotaxis were analyzed as previously described.

### 4.9. CD44 Knock-down

Transfection with small interfering RNA (siRNA) was carried out directed against CD44 (gene ID: 960, target sequence: AACTCCATCTGTGCAGCAAAC, Qiagen, Hilden, Germany). 3 × 10^5^ cells were pre-incubated for 24 h with 2.5 µM SFN and subsequently with a transfection solution of siRNA and transfection reagent (HiPerFect Transfection Reagent; Qiagen) at a ratio of 1:6. Non-treated cells and cells treated with 5 nM control siRNA (AllStars Negative Control siRNA; Qiagen) served as controls. Protein expression, tumor cell adhesion, and chemotaxis were then analyzed as described above.

### 4.10. Statistics

Mean +/- standard deviation was calculated. To exclude coincidence, all experiments were repeated three to five times. Statistical significance was evaluated with the Wilcoxon‒Mann‒Whitney U test and Student’s *t*-test. A *P* value of 0.05 or less indicated a significant difference.

## 5. Conclusions

The present findings reveal that chronic use of the mTOR inhibitor everolimus is associated with drug non-responsiveness, resulting in aggressive migratory activity of bladder cancer cells. Resistance induction caused by everolimus could be inhibited by pairing everolimus with the natural HDAC inhibitor sulforaphane. Patients with bladder cell carcinoma may, therefore, benefit from an anti-tumor strategy including sulforaphane as a complementary component to everolimus.

## Figures and Tables

**Figure 1 ijms-21-05582-f001:**
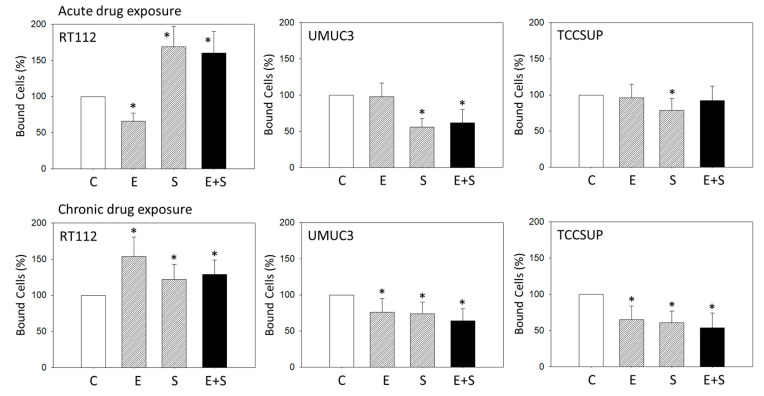
Adhesion of RT112, UMUC3, and TCCSUP cells exposed to 0.5 nM everolimus (E), 2.5 µM sulforaphane (S), or 0.5 nM everolimus + 2.5 µM sulforaphane (E + S). Control cells (C) remained unexposed (set to 100%). Mean number of adherent tumor cells from five fields after 1 h incubation. Bars indicate standard deviation, * indicates significant difference to corresponding control, *p* ≤ 0.05. n = 5.

**Figure 2 ijms-21-05582-f002:**
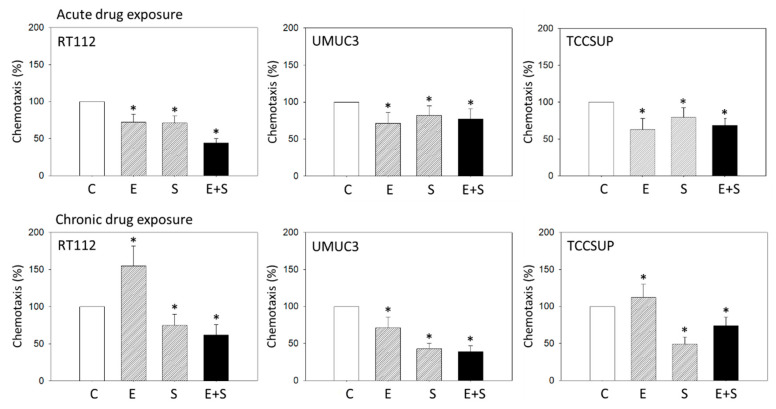
Chemotaxis of RT112, UMUC3, and TCCSUP cells exposed to 0.5 nM everolimus (E), 2.5 µM sulforaphane (S), or 0.5 nM everolimus + 2.5 µM sulforaphane (E + S). Control cells (C) remained unexposed (set to 100%). Mean number of migrating tumor cells from five fields after 24 h incubation. Bars indicate standard deviation, * indicates significant difference to corresponding control, *p* ≤ 0.05. n = 5.

**Figure 3 ijms-21-05582-f003:**
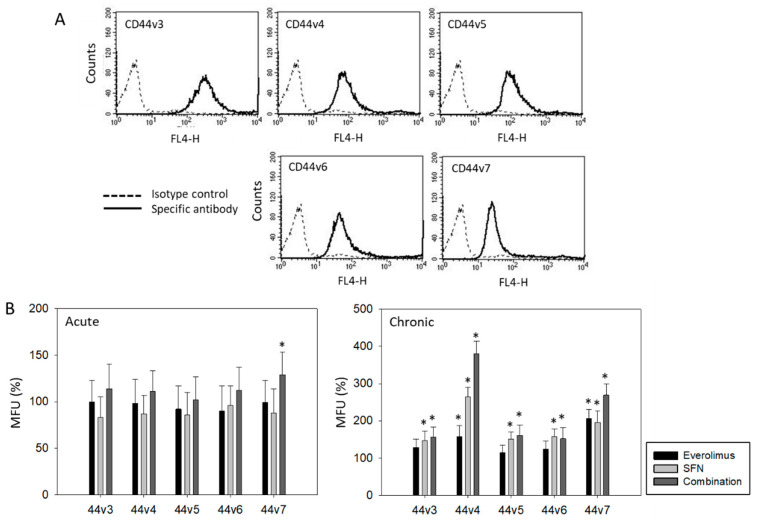
(**A**) Surface expression of CD44 variants v3-v7 on RT112 cells. Counts indicate MFU (mean fluorescence units). One representative of three separate experiments is shown. Solid line: specific fluorescence; dashed line: isotype IgG1-APC. (**B**) expression level of CD44 variants v3-v7 following sulforaphane (SFN) or everolimus + sulforaphane (combination) exposure. All values are means related to untreated controls set to 100% and bars indicate standard deviation. * indicates significant difference to corresponding control, *p* ≤ 0.05, n = 4.

**Figure 4 ijms-21-05582-f004:**
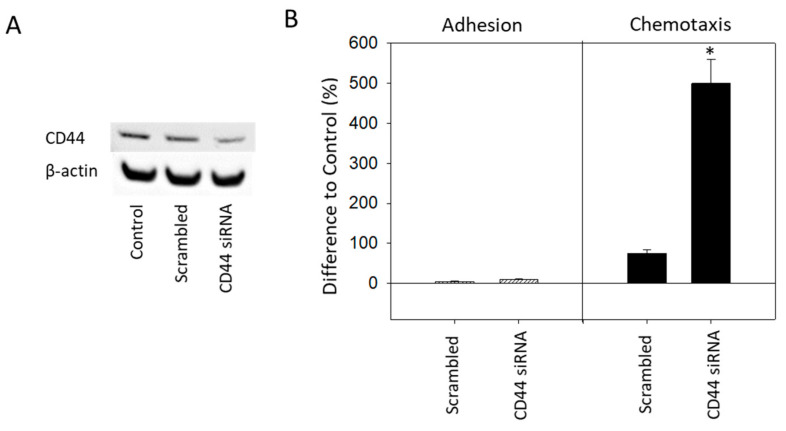
Functional knock-down of CD44 in untreated RT112 cells (Control). AllStars Negative Control siRNA served as transfection control (Scrambled). (**A**) Protein expression after functional blocking with siRNA targeting CD44. β-actin served as internal control. One representative of three separate experiments is shown. (**B**) Tumor cell adhesion and chemotaxis of CD44-blocked RT112 cells. Means ± SD of n = 3. * indicates significant difference to control, *p* ≤ 0.05.

**Figure 5 ijms-21-05582-f005:**
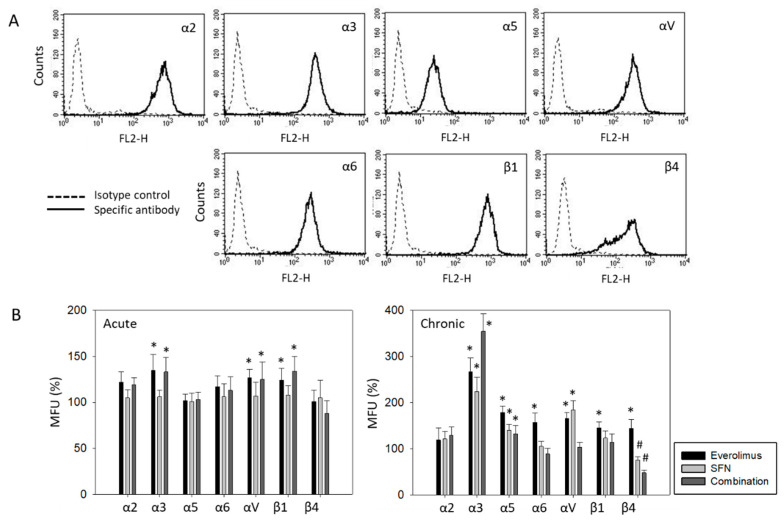
(**A**) Surface expression of integrin α and β subtypes on RT112 cells. Counts indicate mean fluorescence units (MFU). One representative of three separate experiments is shown. Solid line: specific fluorescence; dashed line: isotype control. (**B**) Expression level of integrin α and β subtypes following sulforaphane (SFN), everolimus, or everolimus+sulforaphane (combination) exposure. All values are related to untreated controls set to 100%. Means ± SD of n = 4. * indicates significant difference to corresponding control, # indicates significant difference to everolimus treatment, *p* ≤ 0.05.

**Figure 6 ijms-21-05582-f006:**
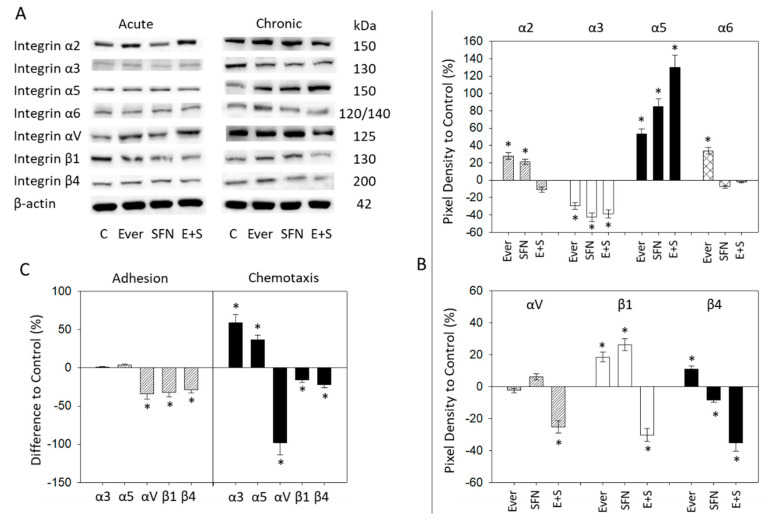
(**A**) Integrin α and β protein analysis in RT112 cells after acute and chronic exposure to sulforaphane (SFN), everolimus (Ever) or everolimus+sulforaphane combination (E+S). β-actin served as internal control. One representative of three separate experiments is shown. Controls (**C**) remained untreated. (**B**) Pixel density analysis (chronic treatment). The ratio of protein intensity/β-actin intensity is expressed as percentage of controls, set to 100%. (**C**) Adhesion and chemotaxis of RT112 cells following functional blocking of integrin α and β subtypes (white columns show everolimus-resistant tumor cells). Means ± SD of n = 3. * indicates significant difference to control, *p* ≤ 0.05.

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
