# Peer review of "Bladder Cancer Metastasis Induced by Chronic Everolimus Application Can Be Counteracted by Sulforaphane In Vitro"

_ijms, 2020, doi:10.3390/ijms21155582_

Round 1

Reviewer 1 Report

In this manuscript "BLADDER CANCER METASTASIS INDUCED BY CHRONIC EVEROLIMUS APPLICATION BLADDER CANCER METASTASIS INDUCED BY CHRONIC EVEROLIMUS APPLICATION CAN BE COUNTERACTED BY SULFORAPHANE IN VITROCAN BE COUNTERACTED BY SULFORAPHANE IN VITRO" the authors reported the use of sulphoraphane as an inhibitor of the resistance-related tumor dissemination during everolimus based bladder cancer treatment.

The manuscript was well written.

The introduction covered a wide range of literature, justifying the study and highlighting the importance if this field in cancer research.

Methods are well described; the results are well described and fit with the purpose of the study.

I think that the manuscript is suitable for publictaion.

Revise english all over the text and please attach images of high resolution quality.

Author Response

The manuscript text has been corrected by an English native speaker. High quality images have been included into the manuscript text. Should further improvement be required during the printing process, we are prepared to further modify the images.

Reviewer 2 Report

The authors demonstrated that chronic exposure to Ever increased cell adhesion in RT112 cells and invasion in RT112 and TCCSUP. The chronic Ever-induced adhesion and invasion were inhibited by the combinatory use of SFN. The authors also demonstrated that adhesion and invasion of RT112 cells were dependent on expression of aV, b1 and b4 integrin, which were downregulated by the combinatory use of SFN with chronic Ever exposure.

The authors may want to show whether the similar modification of integrin expression is observed in TCCSUP (and UMUC3). The authors need to clarify whether the authors' finding is generally observed or not in bladder cancer cells.

The authors did not show what mechanism is directly involved the inhibitory effects of SFN in the authors experimental models. The authors may want to show, at least, blockade of some integrins directly inhibits chronic Ever-induced cell adhesion and invasion.

As described in the discussion section, Akt signaling is important for regulation of cell invasion. The authors may want to provide data on Akt activation after exposure to chronic Ever with and without combination of SFN.

Author Response

Comment 1: The authors may want to show whether the similar modification of integrin expression is observed in TCCSUP (and UMUC3). The authors need to clarify whether the authors' finding is generally observed or not in bladder cancer cells.

Our answer: We have also evaluated integrin expression in TCCSUP and UMUC3 cells following drug exposure. Concerning resistance, similarities between RT112 and TCCSUP were seen in as much SFN elevated α3, αv, and β1 in both cell lines. The α5 subtype was elevated in both cell lines by SFN as well, however, this was only significant in TCCSUP. A different response was evoked in UMUC3 cells. Here, αv was down-regulated by SFN and the integrins α3 and β1 were not altered at all by SFN.

The disparate influence of SFN on several bladder cancer cell lines is not surprising. We demonstrated in an earlier publication that the HDAC–inhibitor valproic acid suppresses adhesion in a broad panel of bladder cancer cell lines by altering integrin α and β expression in a different manner (Juengel E, Meyer dos Santos S, Schneider T, Makarevic J, Hudak L, Bartsch G, Haferkamp A, Wiesner C, Blaheta RA. HDAC inhibition suppresses bladder cancer cell adhesion to collagen under flow conditions. Exp Biol Med (Maywood) 2013; 238: 1297-1304.). The present investigation points to a characteristic receptor set and a non-homogenous integrin pattern with TCCSUP/RT112 compared to UMUC3. The integrin β4 is present on TCCSUP and RT112 but not on UMUC3. On the other hand, β3 has been detected on UMUC3 but not on TCCSUP and RT112. Consequently, it may be expected that drug treatment influences integrin subfamilies in disparate cell lines differently. Indeed, differing integrin guided adhesive behavior in several tumor sublines has been reported. Blocking the α3 integrin subunit inhibited HCV29 bladder cancer cell attachment to the matrix proteins laminin and fibronectin but had an opposite effect on T24 and Hu456 cell adhesion. Similarly, blocking α5 integrin has been shown to down-regulate HCV29 and BC3726 cell-matrix interaction, whereas binding of the bladder cancer cell lines T24 and Hu456 was enhanced (LityÅ„ska A, PrzybyÅ‚o M, Pocheć E, Laidler P. Adhesion properties of human bladder cell lines with extracellular matrix components: the role of integrins and glycosylation. Acta Biochim Pol 2002; 49: 643-650.). In regard to the referee’s comment we evaluated integrin subtype expression on UMUC3 and TCCSUP and analyzed integrin alterations caused by the drug treatment. To avoid overloading the manuscript or making it difficult to follow the main message, we have included these results as supplemental data. The discussion section has been modified as follows (line 270):

“In support, integrin blockage revealed no influence of α3 and α5 on adhesion, pointing to (at least) αV, β1, and β4 as the relevant drivers of resistance in this tumor type.

Investigative results presented here are related to the bladder cancer cell line RT112 and may not be generalizable. Similarities between everolimus-resistant RT112 and TCCSUP cells were seen in as much as SFN elevated α3, αv, and β1 in both cell lines. The α5 subtype was elevated in both cell lines by SFN as well. However, this was only significant in TCCSUP cells. In contrast, a different response was evoked in UMUC3 cells. Here, αv was down-regulated by SFN and the integrins α3 and β1 were not altered (supplemental data). The disparate mechanistic influence of SFN on different bladder cancer cell lines is not surprising. It has previously been shown that the HDAC–inhibitor, valproic acid, suppresses adhesion in a broad panel of bladder cancer cell lines by altering both integrin α and β expression differently [36]. The molecular response of a particular tumor subline seems to depend on the integrin receptor set present on the cell surface. Integrin β4 is present on TCCSUP and RT112 but not on UMUC3 cells, whereas β3 has been detected on UMUC3 but not on TCCSUP and RT112 (supplemental data). Consequently, it may be expected that SFN treatment influences integrin subfamilies and integrin related signaling in different cell lines differently. Indeed, differing integrin guided adhesive behavior in several tumor sublines has been reported. Blocking the α3 integrin subunit has been shown to inhibit HCV29 bladder cancer cell attachment to the matrix proteins laminin and fibronectin but to exert an opposite effect on T24 and Hu456 cell adhesion. Similarly, blocking α5 integrin has been shown to down-regulate HCV29 and BC3726 cell-matrix interaction, whereas binding of the bladder cancer cell lines T24 and Hu456 is enhanced [37]. Based on the current data, it may be assumed that SFN acts on a set of integrin receptors, whereby the integrins modified by SFN may differ according to the initial characteristic integrin composition of the particular cell type.”

Comment 2: The authors did not show what mechanism is directly involved the inhibitory effects of SFN in the authors’ experimental models. The authors may want to show, at least, blockade of some integrins directly inhibits chronic Ever-induced cell adhesion and invasion.

Our answer: We have already presented respective data on the influence of integrin α3, α5, αv, β1 and β4 on RT112 adhesion and chemotaxis (please see figure 6C). We now also performed experiments with everolimus-resistant tumor cells, thereby concentrating on the integrin subtypes β1 and β4. These data have now been included into figure 6. The legend of figure 6C reads: “C) Adhesion and chemotaxis of RT112 cells following functional blocking of integrin α and β subtypes (white columns show everolimus-resistant tumor cells)”. (Line 168)

Comment 3: As described in the discussion section, Akt signaling is important for regulation of cell invasion. The authors may want to provide data on Akt activation after exposure to chronic Ever with and without combination of SFN.

Our answer: In the discussion we do point to the down-regulation of Akt by SFN in renal cell cancer cells (reference 21). Meanwhile, data pertaining to the influence of SFN on mTOR-Akt signaling in bladder cancer cells have been published (Justin S, Rutz J, Maxeiner S, Chun FK, Juengel E, Blaheta RA. Chronic Sulforaphane Administration Inhibits Resistance to the mTOR-Inhibitor Everolimus in Bladder Cancer Cells. Int J Mol Sci. 2020 Jun 4;21(11):4026). In this cell culture model, chronic everolimus administration evoked a significant increase of pAkt. Surprisingly, enhanced Akt phosphorylation was also seen in the presence of SFN alone but not in the presence of the everolimus-SFN-combination, which distinctly reduced pAkt, compared to untreated controls. The mTOR complex pRictor that is associated with cell motility has also been investigated. pRictor was elevated by everolimus, but diminished by SFN and the SFN-everolimus-combination. Therefore, we assume that the process of tumor cell invasion may also be suppressed by SFN via pRictor. Whether the elevation of Akt phosphorylation in the presence of SFN, but not in the presence of the drug combination, may indicate early signs of resistance caused by a feedback mechanism is not yet clear. In regard to this we have now added to the discussion (line 216):

“Therefore, the increase of CD44v4 and CD44v7 induced by SFN could be coupled to the down-regulation of FAK and Akt signaling, finally preventing motile crawling of the tumor cells (via β4 – see below). Nevertheless, Akt deserves further investigation. A recent investigation shows that the influence of SFN on mTOR-Akt signaling in bladder cancer leads to a significant increase in Akt phosphorylation (pAkt) in tumor cells following chronic everolimus administration [23]. Surprisingly, enhanced pAkt was also seen in the presence of SFN alone, but not in the presence of the everolimus-SFN-combination where pAkt was reduced, compared to the untreated controls. The mTOR complex pRictor, associated with cell motility, was also investigated and found to be elevated by everolimus but diminished by SFN and the SFN-everolimus-combination. Therefore, it is possible that the process of tumor cell invasion may also be suppressed by SFN via pRictor. Whether the elevation of pAkt in the presence of SFN indicates early signs of resistance caused by a feedback mechanism is not yet clear”.

Round 2

Reviewer 2 Report

The authors addressed most of the reviewer's comments.